# Emergency Surgery for Colon Diseases in Elderly Patients—Analysis of Complications, and Postoperative Course

**DOI:** 10.3390/medicina58081062

**Published:** 2022-08-06

**Authors:** Mario Pacilli, Alberto Fersini, Giovanna Pavone, Pasquale Cianci, Antonio Ambrosi, Nicola Tartaglia

**Affiliations:** 1Department of Medical and Surgical Sciences, University of Foggia, Luigi Pinto Street, No. 1, 71122 Foggia, Italy; 2Department of Surgery and Traumatology, Hospital Lorenzo Bonomo, 76123 Andria, Italy

**Keywords:** emergency surgery, colon diseases, elderly patients, CR-POSSUM score, postoperative course

## Abstract

***Background and Objectives*:** Colon diseases can turn in a clinical emergency with the onset of some important complications. Some critical conditions are more common in aged patients because they are frailer. The aim of this study is to examine patients over 80 years of age who are undergoing emergency colorectal surgery, and evaluating the aspects associated with post-operative complications and other problems in the short term. ***Methods*:** From November 2020 to February 2022, we included 32 consecutive patients older than 80 undergoing emergency surgery due to colon diseases. We collected and analysed all demographic and operative data, and then applied CR-POSSUM score and correlated this with postoperative hospital stay and the onset of postoperative complications according to the Clavien Dindo classification. ***Results:*** Postoperative factors were selectively evaluated based on the clinical scenario and different colic pathologies. There were no statistically significant differences, in terms of postoperative hospital stay, postoperative complications, reoperation rate and 30-day mortality. The number of cases of blood transfusions was significant and was more numerous in cases of intestinal perforation and bleeding cases. The value of the Operative Severity Score in bowel perforations was significantly higher. ***Conclusions*:** The use of a score to stratify the risk is a useful tool, especially in elderly patients undergoing emergency surgery. The CR-POSSUM score was important for predicting morbidity in our study. Emergency manifestations of colon diseases in the elderly show higher morbidity and mortality rates. The effect of age on outcome is a concept that needs to be emphasized, so further investigation is needed.

## 1. Introduction

Colorectal emergencies are among the most frequent in the field of abdominal surgery [1]. Colon perforation can cause severe sepsis, with subsequent multiple-organ dysfunction, but colonic intestinal occlusions or haemorrhages are also clinical scenarios that should not be underestimated, as they might lead to life-threatening conditions for the patient [2]. Frequently, emergency surgery affects elderly patients; in fact, several critical conditions are more common in geriatric patients because they are frailer and might suffer a delay in diagnosis [3,4].

Surgical choices are mutable and may include treatments with radical or palliative intent, based on the severity of the disease and with the sole intent of saving the patient’s life. The onset of complications such as occlusion, bleeding, and perforation worsen morbidity and postoperative mortality [5,6].

Emergency surgery, in general, is characterised by a higher morbidity and mortality rate in comparison with elective surgery (it can reach rates of 33.6–64% and 20–34%, respectively [7]). Moreover, old age is considered a risk factor for emergency surgery in patients with colorectal diseases, and this assumption often influences the idea that surgery is associated with postoperative risks overwhelming the benefits [8]. These considerations make emergencies associated with colorectal pathologies a real challenge for surgeons.

The aims of this study are to analyse the profile of patients over 80 years of age and undergoing emergency colorectal surgery, evaluating the possible correlation between the factors associated with post-operative complications and other problems in the short term.

## 2. Methods

From November 2020 to February 2022, we performed 253 procedures of colonic surgery. The average age of the total number of patients treated was 69.4 years, with the percentage of patients over 80 at26.8%. A total of 61% of the operations were performed as elective surgery while 39% were performed as an emergency, of which 78 were performed in the first 24 h. All cases of trauma were excluded. A total of 32 consecutive patients older than 80 undergoing emergency surgery (within 24 h) due to colon diseases were included in this retrospective study.

All the demographic data have been collected (Table 1), and in order to better evaluate the factors and the aspects influencing the postoperative course, we separately considered the clinical picture of the emergency and the nature of the disease, collecting data on postoperative hospitalization, onset of postoperative complications (according to the Clavien Dindo classification [9]), blood transfusions, reoperation rate and 30-day mortality, analysed with significance tests and an ANOVA test. We then applied the modified POSSUM score (Physiological and Operative Severity Score for the enUmeration of Mortality and morbidity) for colorectal surgery (CR-POSSUM score [10]) and correlated this with postoperative hospital stay and the onset of postoperative complications according to Clavien Dindo classification, using the Pearson Correlation Coefficient (ρ).

All patients underwent short-term follow-up for up to 30 days after discharge.

Informed consent was obtained from all study participants.

The study was registered on Clinicaltrial with the following ID: NCT05443386.

## 3. Results

Postoperative factors were selectively evaluated based on the clinical scenario determining the surgical emergency (Table 2).

There was no statistically significant difference between the three groups (bowel obstruction, bowel perforation and bleeding), in terms of postoperative hospital stay, postoperative complications, reoperation rate and 30-day mortality. On the other hand, the number of cases of blood transfusions was significant, and was more numerous in cases of intestinal perforation and bleeding (Table 2).

No significant differences were found in the three groups of colic pathologies considered (malignancy, diverticulitis, ischemic bowel disease). The ischemic bowel disease group had an average postoperative hospital stay; the stay was longer in cases of blood transfusions compared to the other two groups, albeit not significantly longer (Table 3).

All patients in the study underwent the CR-POSSUM score. We selectively analysed the three parameters of the CR-POSSUM (physiology score, operative severity score, mortality), for each clinical scenario and colon disease. In our cohort, the highest value of Operative Severity Score in the bowel perforations group was statistically significant (Table 4 and Table 5).

Finally, we analysed the results obtained with Pearson’s Tests of Correlation. A significant correlation was observed between physiology score, mortality rate and postoperative complications according to the Clavien Dindo classification (ρ = 0.479 *p* value = 0.05; ρ = 0.399 *p* value = 0.023). In the analysis of the postoperative hospital stay, there was always a positive correlation (ρ > 0) but this was not statistically significant (Figure 1 and Figure 2).

## 4. Discussion

Acute abdomen in elderly patients represents a real challenge to all surgeons. An atypical presentation of the disease occurs very frequently in this group of patients and the diagnosis is often only possible by employing instrumental tests. The diagnosis might also come late, as a consequence of these patients living in alienation and not adhering to screening programs, and these aspects are further accentuated in the pandemic SARS-CoV-2 era [11,12]. If the presence of a surgical condition is confirmed, then surgical treatment is mandatory and the decision making in such cases might be very challenging.

The results of our study show that elderly patients undergoing emergency surgery for colonic disease suffer from a 30-day mortality rate of 15.6% and a malignant disease rate of 25%.

In our study, documented complications were well controlled, with only four cases (12.5%) of reoperations (Clavien Dindo IIIa). Many data obtained between the comparisons of the three groups are not statistically significant, although we recorded an unfavourable postoperative course for intestinal perforations, among the different clinical scenarios, and for intestinal ischemia when comparing colon diseases. In the evaluation of the CR-POSSUM score, the difference in the Operative Severity Score compared between the different clinical scenarios was significant (Table 4).

The principal emergencies related to colorectal diseases are intestinal obstruction, haemorrhage and perforation. Intestinal obstruction is the most frequent in the literature [13,14] and represents 46.8% of the cases in our study. The cases of malignancy are often characterized by advanced stage disease and/or metastatic disease, for which surgery is often not performed with radical intents but with the aim of saving the patient’s life. This circumstance occurs more frequently in elderly patients [15,16].

Age cannot represent a prognostic factor by itself when dealing with survival rates in colonic surgery [17], but there is no doubt that the presence of comorbidities in elderly patients is higher and influence the clinical course [18].

The high mortality and morbidity rates described in emergency interventions emphasise the need for vigilant preoperative assessments to correct comorbidities, as colon diseases also lead to special deficit states such as anaemia, malnutrition and sepsis in the worse scenarios [19,20].

From these premises arise the need to categorize these patients according to a reliable scoring system, which might allow us to objectively stratify the perioperative risk and better communicate with the patients’ relatives. CR-POSSUM comes from the POSSUM and *p*-POSSUM models for surgical mortality and morbidity risk estimate. This adjustment of the POSSUM model is indicated for patients undergoing colorectal surgery [21]. It can be employed for both emergency and elective surgery, but there is not a specific value indicating emergency treatment, and this is, in our opinion, a limitation of this score. On the other hand, a worse score will be assigned considering the patient’s age, advanced cancer stage or in cases of peritoneal contamination.

In our study, therefore, we thought that CR-POSSUM, originally developed to predict mortality, might also be utilized to foresee morbidity. To demonstrate this, we performed an analysis using the Pearson Correlation Coefficient between the values obtained in the CR-POSSUM, and the postoperative hospital stay and staged complications according to the Clavien Dindo classification. We intend for the postoperative hospital stay value to be used as a parameter that could correlate with the recovery of the elderly and frail patient, as well as being a parameter related to general costs [22]. We consider the obtained results interesting because, in all cases, a positive relationship with a ρ > 0 (except for Operative Severity Score and the Clavien Dindo Classification) was recorded and the acquired data were statistically significant when compared with Physiology Score and the Clavien Dindo Classification (ρ = 0.479 *p* value = 0.05), and Mortality Rate and the Clavien Dindo Classification (ρ = 0.399 *p* value = 0.023). We could infer that the values obtained from the CR-POSSUM score can be considered separately and that this approach can help surgeons in predicting the clinical trends of the patient operated on in an emergency.

## 5. Conclusions

Dealing with the stratification of operative risk, we believe in the usefulness of a score that provides objective information and allows the assessment of mortality and morbidity risk values. This might also be considered a useful tool in communicating with the patients, who might only rely on surgery as a therapeutic choice. The CR-POSSUM score is a consistent scoring system that expresses a significant value in predicting morbidity, although no indicator assigning a higher score to emergency interventions is available.

Emergency manifestations of colon diseases in the elderly are frequent and show higher morbidity and mortality rates in comparison with elective admissions. The impact of old age on outcomes is a concept that must be underlined and emphasized, because it is associated with poor surgical results in this population.

In critical situations, the patient’s survival must be the only goal to pursue, despite the possibility of a complicated postoperative course. Further investigation with adjunctive prospective studies will improve our knowledge on these situations.

## Figures and Tables

**Figure 1 medicina-58-01062-f001:**
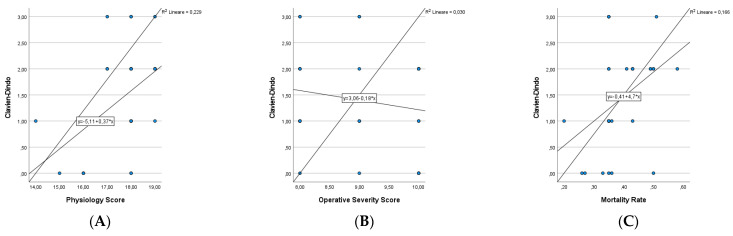
Pearson correlation coefficient to predict postoperative hospital stay. (**A**) Physiology Score and Postoperative Hospital Stay (ρ = 0.193 *p* value = 0.289). (**B**) Operative Severity Score and Postoperative Hospital Stay (ρ = 0.07 *p* value = 0.703). (**C**) Mortality Rate and Postoperative Hospital Stay (ρ = 0.269 *p* value = 0.136).

**Figure 2 medicina-58-01062-f002:**
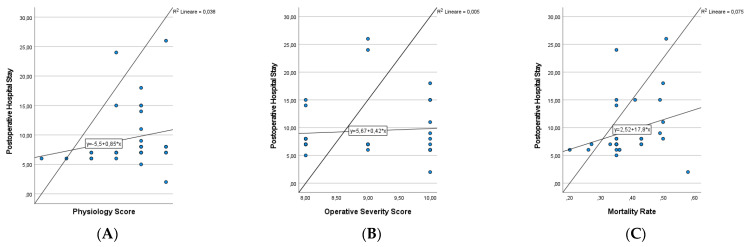
Pearson correlation coefficient to predict postoperative complications according to Clavien Dindo Classification. (**A**) Physiology Score and Clavien Dindo Classification (ρ = 0.479 *p* value = 0.05). (**B**) Operative Severity Score and Clavien Dindo Classification (ρ = −0.174 *p* value = 0.340). (**C**) Mortality Rate and and Clavien Dindo Classification (ρ = 0.399 *p* value = 0.023).

**Table 1 medicina-58-01062-t001:** Demographic data. Patients treated in emergency surgery for colon diseases. Data are expressed as mean, or percentage values.

	*n* = 32
Age (years)	86 (80–95)
Male sex	16 (50%)
BMI	24.5
Previous surgery	38%
Malignancy Rate	25%
Death at 30 days	15.6%

**Table 2 medicina-58-01062-t002:** Postsurgical data in different clinical pictures. Significance tests used are ANOVA and Chi-Square.

	Bowel Obstruction (n = 15)	Bowel Perforation (n = 11)	Bleeding (n = 6)	
Postoperative Hospital Stay	8.3 (7–15)	11.1 (2–26)	9 (5–15)	*p* = 0.434
Clavien Dindo				
Grade 0	2/15 (13.3%)	3/11 (27.3%)	1/6 (16.7%)	
Grade I	7/15 (46.7%)	2/11 (18.2%)	2/6 (33.3%)	
Grade II	5/15 (33.3%)	4/11 (36.4%)	2/6 (33.3%)	
Grade IIIa	1/15 (6.7%)	2/11 (18.2%)	1/6 (16.7%)	*p* = 0.812
Blood Transfusion	2/15 (13.3%)	4/11 (36.4%)	4/6 (66.7%)	*p* = 0.05
Reoperation	1/15 (6.7%)	2/11 (18.2%)	1/6 (16.7%)	*p* = 0.641
30 days Mortality	3/15 (20%)	2/11 (18.2%)	0	*p* = 0.907

**Table 3 medicina-58-01062-t003:** Postsurgical data in different colon diseases. Significance tests used are ANOVA and Chi-Square.

	Malignancy (n = 12)	Diverticulitis (n = 15)	Ischemic Bowel Disease (n = 5)	
Postoperative Hospital Stay	9.5 (7–15)	8.6 (2–24)	11.6 (7–26)	*p* = 0.567
Clavien Dindo				
Grade 0	1/12 (8.3%)	4/15 (26.7%)	1/5 (20%)	
Grade I	5/12 (41.7%)	5/15 (33.3%)	1/5 (20%)	
Grade II	4/12 (33.3%)	5/15 (33.3%)	2/5 (40%)	
Grade IIIa	2/12 (16.7%)	1/15 (6.7%)	1/5 (20%)	*p* = 0.863
Blood Transfusion	4/12 (33.3%)	3/15 (20%)	3/5 (60%)	*p* = 0.242
Reoperation	2/12 (16.7%)	1/15 (6.7%)	1/5 (20%)	*p* = 0.633
30 days Mortality	3/12 (25.0%)	1/15 (6.7%)	1/5 (20%)	*p* = 0.409

**Table 4 medicina-58-01062-t004:** CR-POSSUM in different clinical pictures. Significance test used is ANOVA test.

	Bowel Obstruction (n = 15)	Bowel Perforation (n = 11)	Bleeding (n = 6)	*p* Value
CR-POSSUM				
Physiology Score	17.866	17.091	17.5	0.283
Operative Severity Score	8.533	9.818	8.5	<0.001
Mortality (%)	37.9%	42.1%	35%	0.191

**Table 5 medicina-58-01062-t005:** CR-POSSUM in different colon diseases. Significance test used is ANOVA test.

	Malignancy (n = 12)	Diverticulitis (n = 15)	Ischemic Bowel Disease (n = 5)	*p* Value
CR-POSSUM				
Physiology Score	17.416	17.467	18	0.655
Operative Severity Score	8.833	9.133	8.8	0.635
Mortality (%)	36.6%	39.8%	41.1%	0.488

## Data Availability

All data generated or analysed during this study are included in this article. Further enquiries can be directed to the corresponding author.

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
