# Peer review of "Emergency Surgery for Colon Diseases in Elderly Patients—Analysis of Complications, and Postoperative Course"

_medicina, 2022, doi:10.3390/medicina58081062_

Round 1

Reviewer 1 Report

Comments to Author,

1. Axis of graphs 1 and 2 not clear and use some software to draw the graph

2. Provide some molecular data that will increase the scientific sound of the manuscript 

3. Provide the Statistical analysis information in methodology  

Reviewer 2 Report

Letter to authors:

 Comments

In this article the authors aim to address a relevant clinical complication observed in the elderly above the age of 80 with bowel abnormalities. The results of the study show that elderly patients undergoing emergency surgery for colonic disease suffer from a 30-day mortality rate of 15.6% and a malignant disease rate of 25%.

Major comments:

-          In the methods section of page 1, the authors state that – “We have collected and analysed all demographic, and operative data and then applied CR-POSSUM score and correlated with postoperative hospital stay and the onset of postoperative complications according to Clavien Dindo classification.” The authors do not address the “onset” issue anywhere in the paper. This needs to be properly discussed and is a major concern.

-          On page 3, the authors mention that – “In our cohort, the highest value of Operative Severity Score in the Bowel Perforations group was statistically significant (Table 4 and Table 5).” Yet in Table 5, the CR-POSSUM scores indicate that the p-value is much larger (0.635). Why do the authors say this?

-          The authors need to work and add more details to the Graphs 1 and 2 in terms of the adding a trendline, r2 and p-values in the graph itself. Also, the findings are mostly based on correlations which are albeit positive, yet not strong enough. So the claims should be tender.

Minor comments:

-         Reword the 3rd sentence in the results section - In our study the complications recorded they were easily manageable, with only four cases (12.5%) of reoperations (Clavien Dindo IIIa).

-          Grammatical errors in multiple places for example – “Intestinal obstruction is the most frequent in literature [13,14] and represents 46.8% of the cases in ours study.”

-          Correct for gramma – “Many data obtained between the com-
parisons of the three groups are not statistically significant, although we recorded a unfavourable postoperative course for intestinal perforations, among the different clinical scenarios, and for intestinal ischemia when comparing colon diseases.”

Round 2

Reviewer 1 Report

The author has revised the manuscript in good manner and accepted for publication. 

Author Response

Response to Reviewer 1 Comments

Point 1:The author has revised the manuscript in good manner and accepted for publication. 

Response 1: Dear reviewer,
thank you very much for your kind cooperation

Reviewer 2 Report

The authors fail to underscore how their model (CR-POSSUM score) correlates to the onset of post-operative complications

Author Response

Response to Reviewer 1 Comments

Point 1: The authors fail to underscore how their model (CR-POSSUM score) correlates to the onset of post-operative complications

Response 1: Dear Reviewer, Thanks again for the suggestions, which are useful to improve our work.

A correction was made at line 157 using the Track Changes function. We have dealt with the correlation between the CR POSSUM score and the onset of complications, in the Discussion section from line 182 to line 192. However, we have implemented the text to better underline the correlation between the CR POSSUM score and the onset of complications.